# Spontaneous Calcium Bursts Organize the Apical Actin Cytoskeleton of Multiciliated Cells

**DOI:** 10.3390/ijms26062507

**Published:** 2025-03-11

**Authors:** Johannes Wiegel, Martin Helmstädter, Gerd Walz, Max D. Bergen

**Affiliations:** 1Department of Medicine IV, University Freiburg Medical Center, Faculty of Medicine, University of Freiburg, Hugstetter Strasse 55, 79106 Freiburg, Germany; johannes.wiegel@uniklinik-freiburg.de (J.W.); martin.helmstaedter@uniklinik-freiburg.de (M.H.); gerd.walz@uniklinik-freiburg.de (G.W.); 2EMcore, Renal Division, Department of Medicine, University Freiburg Medical Center, Faculty of Medicine, University of Freiburg, 79106 Freiburg, Germany; 3BIOSS and CIBSS Centre for Integrative Biological Signalling, University of Freiburg, Schänzlestrasse 18, 79104 Freiburg, Germany

**Keywords:** calcium signaling, actin, cytoskeleton, RhoA, cilia, optogenetics, *Xenopus*

## Abstract

Motile cilia perform crucial functions during embryonic development and in adult tissues. They are anchored by an apical actin network that forms microridge-like structures on the surface of multiciliated cells. Using *Xenopus* as a model system to investigate the mechanisms underlying the formation of these specialized actin structures, we observed stochastic bursts of intracellular calcium concentration in developing multiciliated cells. Through optogenetic manipulation of calcium signaling, we found that individual calcium bursts triggered the fusion and extension of actin structures by activating non-muscle myosin. Repeated cycles of calcium activation promoted assembly and coherence of the maturing apical actin network. Inhibition of the endogenous inositol triphosphate-calcium pathway disrupted the formation of apical actin/microridge-like structures by reducing local centriolar RhoA signaling. This disruption was rescued by transient expression of constitutively active RhoA in multiciliated cells. Our findings identify repetitive calcium bursts as a driving force that promotes the self-organization of the highly specialized actin cytoskeleton of multiciliated cells.

## 1. Introduction

Motile cilia perform crucial functions in the human body, including establishing left–right asymmetry, clearing the airways of mucus, promoting cerebrospinal fluid circulation, and transporting oocytes in the fallopian tubes [1,2]. Defects in ciliary motility can lead to severe conditions known as motile ciliopathies [3]. The motile cilia of the epithelia in the airways, ependyma, and reproductive tract extend from the apical surface of multiciliated cells (MCCs), a highly specialized cell type with up to 300 cilia per cell [1,3]. MCCs also occur in the embryonic epidermis of post-neurula *Xenopus* embryos and tadpoles, sharing a conserved process of multiciliogenesis with their mammalian counterparts, thus providing an excellent model for studying mucociliary epithelia [4]. 

In *Xenopus* embryos, MCCs differentiate in the inner epidermal cell layer [2], where they generate a large number of centrioles that serve as basal bodies of motile cilia [5]. After intercalation into the outer layer, the surface area of MCCs expands as basal bodies continue to dock to the apical plasma membrane. This process is essential for the generation and extension of the ciliary axoneme [1]. Both surface expansion and basal body docking depend on the actin cytoskeleton and actin modulators such as Rho GTPases and Ezrin [6,7,8]. RhoA-driven actin polymerization provides the pushing force for apical expansion [9,10], while active RhoA recruited to basal bodies is necessary for their apical migration. RhoA recruitment requires components of the planar cell polarity (PCP) pathway, including Dishevelled, Inturned, Daam1, and Drg1, as well as the ciliary protein Ccdc108 [11,12,13,14]. As more basal bodies dock to the apical surface, the apical actin cytoskeleton organizes around the basal bodies to form a tight network [15], providing stability against the forces exerted by ciliary beating [16]. However, the precise mechanisms by which actin fibers form and organize into a network, and how cells regulate this process, remain unclear. Basal bodies become anchored to the apical actin network by ciliary adhesions at their appendages, known as rootlets and basal feet [6,17,18]. We recently demonstrated that the apical actin network generates three-dimensional structures on the surface of MCCs, closely resembling microridges [18]. Microridges are elongated protrusions on epithelial surfaces, composed of branched actin and keratin filaments, and their formation requires non-muscle myosin II (NMII) contraction [19,20,21,22,23]. NMII regulation and activity are typically controlled by calcium ions, which bind to calmodulin and activate myosin light chain kinase (MLCK), subsequently activating NMII through light chain phosphorylation [24]. In mature MCCs, calcium signaling, in combination with cyclic AMP and GMP signaling, regulates ciliary beat frequency [25].

We recently demonstrated that MLCK inhibition impairs the formation of the microridge-like structures on multiciliated cells of the *Xenopus* epidermis [18]. In the present study, we investigated how MCCs control the development of the apical actin network and microridges. We observed stochastic calcium bursts in MCCs during ciliogenesis. Using optogenetics, we showed that a single calcium transient increased the length and interconnectivity of the apical actin network, in an NMII-dependent manner. We also confirmed that long-term stimulation of calcium signaling promotes apical actin network development, while inhibition of NMII disrupts it. Furthermore, investigating the mode of calcium signaling revealed that the inositol trisphosphate-calcium signaling pathway is required for microridge formation by promoting RhoA activity at ciliary basal bodies.

## 2. Results

### 2.1. Calcium Activity in Multiciliated CELLS Drives Apical Actin Network Development

Microridge-like structures on the apical surface of multiciliated cells (MCCs) of the *Xenopus* epidermis consist of a dense apical actin network embedded in the apical plasma membrane and serve to anchor motile cilia. Microridge development in zebrafish epidermal cells requires myosin-driven contractions [22]. To elucidate the regulation of microridge-like structure development in MCCs, we analyzed calcium activity and cortical contraction in MCCs. We used the genetically encoded calcium sensor protein GCaMP6s, which increases in fluorescence in the presence of calcium [26]. We observed frequent calcium transients (bursts) in the developing *Xenopus* mucociliary epithelium. Most bursts occurred in MCCs, with occasional bursts in other intercalating cell types (Figure 1A). 

Although calcium bursts in MCCs were relatively frequent, they appeared stochastically in individual MCCs, which made a comprehensive analysis difficult. We therefore employed the optogenetic BACCS (blue light-activated calcium channel switch [27]) system to induce calcium bursts in MCCs, and monitored responses using confocal time lapse imaging (Figure 1B). The BACCS system uses a calcium channel located in the plasma membrane (Orai) and light-sensitive fusion protein (BACCS) including a peptide from the Orai interactor STIM1. Exposure to blue light causes a conformational change in BACCS, which allows interaction with Orai, opening the calcium channel [27].

Optogenetic stimulation of calcium entry induced an immediate cortical contraction of MCCs, reflected in a reduction in the apical surface. After the calcium burst, cells showed relaxation and re-expansion (Figure 1C). Dynamic changes in the apical actin network followed each calcium burst (Figure 1D,E).

**Figure 1 ijms-26-02507-f001:**
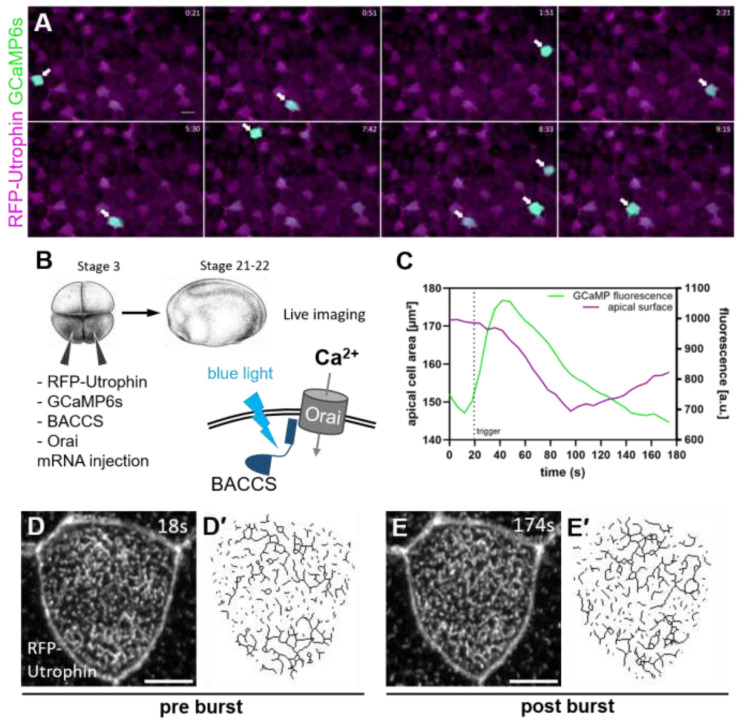
Calcium bursts in MCCs cause apical contraction and changes to the apical actin network: (**A**) Calcium bursts occur regularly in MCCs during the development of the *Xenopus* epidermis (arrows). Stills of a movie recorded over 10 min (20 frames/minute) of a stage 21–22 embryo with actin (RFP-Utrophin, magenta) and calcium (GCaMP6s, green) labels. Scale bar 20 µm. (**B**) Setup for experimental induction of calcium bursts in MCCs. Targeted activation of BACCS by blue light causes Ca^2+^-influx through Orai channels. *Xenopus* illustrations © Natalya Zahn (2022), xenbase.org RRID:SCR_003280 [28]**.** (**C**–**E**) Effects of stimulated calcium bursts on MCCs in a representative cell. (**C**) Graph of GCaMP fluorescence and apical surface of an MCC during an optogenetically induced calcium burst. (**D**,**E**) Maximum intensity projections of apical actin labeled by RFP-Utrophin in a developing MCC before and after an induced calcium burst. (**D’**,**E’**) Skeletonized actin derived from the confocal images. Scale bars 5 µm.

To investigate these changes, we recorded a series of Z-stacks of MCCs over a period of 180 s. After 12 s, we induced a calcium burst by blue laser stimulation. We then quantified and compared the final Z-stack of the series to the one recorded just before the calcium burst (Figure 2A). For quantification, we used an ImageJ plugin to skeletonize confocal images of apical actin (method illustrated in Appendix A). Early observations suggested that actin structures increased in length and/or fused with neighboring actin structures during calcium bursts. Quantification confirmed that the number of actin skeletons (separate actin structures) consistently decreased in MCCs undergoing a calcium burst and apical contraction, but not in MCCs without calcium activity (Figure 2B,E). At the same time, the number of junctions within the actin network increased, while the number of branches remained unchanged (Appendix A), indicating that actin skeletons were indeed fusing. We further observed that the average branch length of the apical actin network increased after a calcium burst, perhaps due to increased actin polymerization (Figure 2B,E). We concluded that a single calcium burst is able to induce significant changes in the apical actin network, driving it towards a more interconnected and mature state. 

### 2.2. Non-Muscle Myosin II Promotes Coherence of the Apical Actin Network

Cellular contraction usually requires myosin activity. Since contraction and actin assembly coincided in our MCC calcium burst model, we suspected that the dynamic changes in the apical actin network could also be myosin-dependent. We thus treated *Xenopus* embryos with NMII inhibitor Blebbistatin before inducing calcium influx using BACCS. NMII inhibition strongly reduced apical surface contraction and the effects on apical actin described above (Figure 2C,E and Appendix A). While the treatment did not completely suppress either contraction or actin skeleton fusion, it significantly reduced both (Figure 2E). However, Blebbistatin did not prevent the increase in branch length (Figure 2E). We then tested whether actin-nucleating formin proteins played a role in the calcium response by using SMIFH2 (small molecule inhibitor of FH2 domain). In contrast to myosin inhibition, this treatment prevented the increase in branch length during calcium bursts but did not affect contraction and fusion of actin skeletons (Figure 2D,E and Appendix A), suggesting that actin polymerization plays a role independent of myosin-based actin remodeling during calcium-induced actin assembly in MCCs.

Based on these results, we hypothesized that MCCs undergo repeated cycles of calcium-mediated contractions, followed by relaxation and cell surface expansion. In these cycles, every calcium burst incrementally increases coherence and length of the apical actin network, involving activation of NMII. We use the term ‘coherence’ here and subsequently to refer to the degree to which the apical actin structures are interconnected.

To test whether Blebbistatin impairs the development of the actin network, we incubated embryos with Blebbistatin throughout the entire apical actin development (embryonic stages 19 to 30). This treatment resulted in very uneven development of the apical actin network (Figure 3A,B). In some areas, actin structures were very dense, while in other areas of the same cell, actin bundles were sparse and fragmented. Notably, the distribution of basal bodies over the apical surface was equally affected. Quantitative analysis of apical actin revealed higher numbers of actin skeletons as well as a less branched and slightly shorter actin network; however, the apical area of MCCs did not change (Figure 3C and Appendix A).

We also tested whether additional calcium bursts during apical actin development would increase actin network length and coherence compared to control MCCs. Embryos were injected with the components of the BACCS system and subjected to blue light pulses every 4 min from stage 19 to stage 30 (Figure 3D). Notably, MCCs in illuminated embryos had fewer actin skeletons with more branches and slightly longer apical actin networks compared to embryos kept in the dark (Figure 3E and Appendix A), suggesting that calcium bursts have a long-term effect promoting apical actin development.

These observations suggest that calcium bursts trigger incremental maturation of the apical actin network in MCCs, involving NMII-based contractions and the fusion of actin bundles. 

### 2.3. Inositol Trisphosphate/Calcium Signaling Is Required for Apical Actin Integrity

We next studied the mode of calcium signaling during MCC development. The endoplasmic reticulum (ER) is the main cellular calcium storage. Intracellular calcium in non-excitable cells is released by the opening of ligand-gated ion channels in the ER membrane in response to inositol trisphosphate (IP_3_) [29]. We treated stage 19 embryos with the IP_3_ receptor inhibitor 2-aminoetoxydiphenyl borate (2-APB) to test the role of IP_3_-dependent calcium signaling in apical actin development. The treatment resulted in severe truncation of the apical actin network into small actin fragments by stage 30 (Figure 4A,B,G–J). Scanning electron microscopy confirmed that 2-APB treatment altered the microridge architecture in the same way it changed the apical actin network (Figure 4D–E).

2-APB also affects store-operated calcium entry (SOCE) in addition to inhibiting IP_3_ receptors (IP3R). To verify that the observed effects were due to IP3R inhibition, we blocked IP_3_ production by phospholipase C (PLC) using U73122 (Figure 4F) and inhibited SOCE with YM58483. Embryos treated with U73122 showed the same MCC apical actin defects as those treated with 2-APB (Figure 4C,K–N), while YM58483 treatment did not affect the cytoskeleton of MCCs (Appendix A). We concluded that the disruption of apical actin was caused specifically by the loss of IP_3_-induced calcium release, rather than toxicity or interference with SOCE.

We previously showed that the microridge-like structures on MCCs play a role in polarizing motile cilia [18]. This suggests that embryos subjected to IP_3_ inhibition would lack polarization of motile cilia and coordinated ciliary beating. We thus assessed cilia polarization by measuring the orientation of the basal body rootlets. Rootlet orientation was disorganized in 2-APB-treated embryos compared to controls (Figure 5A–C). We further quantified the gliding of embryos on agarose as a consequence of epidermal ciliary beating, using it as a measure of MCC function [30]. Gliding was significantly slower or absent in embryos treated with 2-APB or U73122 compared to controls (Figure 5D), indicating that IP_3_ inhibition disrupts ciliary flow.

Establishing and maintaining basal body polarity not only requires the apical and subapical actin networks but also a microtubular network [17]. We labeled microtubules using EMTB-3xGFP to assess the potential effect of IP3R inhibition on this structure, but did not observe any substantial changes (Appendix A). However, the EMTB-3xGFP construct also labels the proximal ciliary axoneme and we observed a striking loss of this label in 2-APB-treated cells (Appendix A). The number of basal bodies remained unchanged. We subsequently used immunofluorescence against acetylated tubulin and α-tubulin and confirmed that IP3R inhibition leads to a moderate loss of cilia in MCCs (Appendix A).

In conclusion, IP_3_-calcium signaling is required for the development of the apical actin network and microridges. The inhibition of IP_3_ impairs actin network formation and basal body polarization, causes loss of cilia, and thereby impairs mucociliary function.

### 2.4. RhoA Acts Downstream of Calcium Controlling Apical Actin Development

Small GTPases of the Rho family are regulators of the actin cytoskeleton, and RhoA plays an established role in MCC surface expansion and apical actin formation [10]. Previous studies have shown that active RhoA is present in close proximity to basal bodies during MCC development [12,14]. We thus asked whether 2-APB treatment interferes with the localization or activation of centriolar RhoA. Using GFP fused to the GTPase-binding domain of Rhotekin (rGBD) to identify RhoA activity [31], we observed that activated RhoA primarily localized to the anterior rootlet, as visualized by mCherry-*clamp* (Figure 6A). When embryos were treated with 2-APB, apical rGBD reporter fluorescence was lower, and localization to the rootlet was disrupted compared to controls (Figure 6A–D). This finding supports a role for calcium in RhoA activation and suggests that RhoA activation occurs downstream of IP_3_-calcium signaling.

To determine whether decreased RhoA activity caused the disruption of apical actin in 2-APB-treated embryos, we used dominant negative (T19N) or constitutively active (Q63L) human RHOA. Targeted expression of RHOA^T19N^ in MCCs led to truncation of the apical actin network, similar to the phenotype observed after disruption of the IP_3_ pathway. Expression of RHOA^Q63L^ did not disrupt apical actin (Figure 6E–J). On the contrary, MCCs expressing RHOA^Q63L^ were no longer susceptible to 2-APB treatment (Figure 7A–D). Skeleton-based quantification of the apical actin network showed that RHOA^Q63L^ almost completely rescued the phenotype caused by 2-APB treatment (Figure 7E–H).

One of the major effectors of RhoA is Rho-associated coiled-coil-containing protein kinase (ROCK). We used the ROCK inhibitor Y27632 to test whether ROCK activity plays a role in MCC apical actin formation. Treating embryos with Y27632 resulted in a truncation of the apical actin network (Appendix A), confirming the central role of RhoA as a downstream target of IP_3_-calcium signaling in MCCs. 

## 3. Discussion

We identified a novel role for calcium during MCC development, driving apical contractions and actin remodeling. MCCs exhibit repetitive calcium transients after intercalation, which coincide with apical contractions. In our model, using the optogenetic BACCS tool to induce calcium bursts, we observed the extension and increased interconnectivity of the apical actin network in individual cells. 

There are reports of calcium-dependent actin remodeling in various cell types. Calcium-induced assembly of nuclear actin filaments in fibroblasts occurs within seconds and depends on calmodulin and the formin protein INF2 [32]. Calcium-dependent activation of INF2 has also been reported to affect mitochondrial fission [33] and to promote actin polymerization during the reset of cortical actin in response to stress and acute signals [34]. In T lymphocytes, T cell receptor activation triggers Arp2/3-dependent actin polymerization via calcium, calmodulin, and N-WASP [35]. Furthermore, the immune synapse of T cells is shaped by actin remodeling under the control of calcium influx through CRAC channels [36]. Similar to our observations in MCCs, calcium influx in these systems leads to rapid polymerization or remodeling of actin fibers on a timescale of seconds to minutes. However, these processes appear to differ significantly in several ways. Firstly, we do not observe substantial formation of new actin structures but rather the extension and fusion of those already present. Secondly, we do not observe the sudden loss of cortical or apical actin structures, as seen in an actin reset [34]. Finally, the calcium response of MCCs is predominantly defined by actin remodeling through myosin activity.

Myosin-dependent assembly of apical actin microridges occurs in epidermal cells of the zebrafish [20]. In these cells, as well as in cultured kidney epithelial cells [37], microridges arise from dot-like apical actin structures termed “pegs”. We observed pegs and very short ridges in early MCCs and at later stages after inhibition of IP3R, ROCK, or other treatments that interfere with the formation of the apical actin network ([18], this study). Pegs coalesce into larger structures over time in an NMII-dependent manner, coinciding with contractions of the apical cell surface [20,22], which is reminiscent of our observations in MCCs. It is unknown, however, whether calcium is involved in the process in the zebrafish epidermis. The coalescence of actin in zebrafish depends on the stochastic action of NMII mini-filaments that fuse, cleave, and rearrange pegs and microridges [23]. A similar mode of NMII action might also be present in *Xenopus* MCCs, as we observe NMII-dependent fusion of actin structures following calcium bursts. Importantly, however, the spatial arrangement of apical microridges differs significantly between zebrafish epidermal cells and MCCs. A potential cause of this difference is the presence of basal bodies in MCCs, which likely contribute to actin organization through actin-stabilizing factors such as RhoA and WDR5 [15]. Notably, we observed that NMII inhibition disrupts the proper distribution of basal bodies, alongside apical actin assembly. This suggests that basal body spacing and actin network development influence each other. A highly connected apical actin network can apparently form in areas of high basal body density even without NMII (Figure 3B), but myosin is required to assemble an evenly spaced actin network surrounding evenly spaced basal bodies.

RhoA is possibly involved in the regulation of NMII, since the Rho-activated protein kinase ROCK can activate NMII by phosphorylating the regulatory light chain. However, our results indicate that one or more additional factors promote the development of the actin network in response to calcium. Formin inhibitor SMIFH2 prevented the increase in actin branch length after calcium bursts. Formins are the most studied actin nucleators and many of them interact with Rho GTPases. This includes Daam1, which has known functions in MCCs [11,13]. The formin INF2 is activated directly by calcium/calmodulin, but it is not localized to the apical actin network [34,38]. The WH2 domain-containing protein Cobl is another actin nucleator regulated by calcium/calmodulin and is known to be required for motile-ciliated cell development, but is not inhibited by SMIFH2 [16,39,40].

It has been observed that SMIFH2 inhibits NMII and several other myosins, in addition to formin FH2 domains [41]. However, the effects of Blebbistatin and SMIFH2 treatments observed by us were quantitatively different, with the concentration of SMIFH2 used being relatively low (10 µM). Therefore, we believe that the effect observed with SMIFH2 treatment is unlikely to be caused by inhibition of NMII, but we cannot exclude this possibility based on our experiments.

We found that RhoA maintains the integrity of the apical actin network during MCC development downstream of calcium. Notably, IP_3_-calcium signaling is also required for motile cilia maintenance in zebrafish ependymal cells [42]. RhoA performs multiple functions during MCC development prior to apical network and microridge formation. Its expression is controlled by major regulators of multiciliogenesis, including Foxj1 and miR-34/449 [8,43]. Localization and activation at basal bodies are required for basal body migration and docking, and depend on Ccdc108, Inturned, and Dishevelled through its regulation of Drg1 and Daam1 [11,12,14]. Immediately after the intercalation of MCCs into the outer epidermal cell layer, apical RhoA activity promotes actin polymerization, which drives apical surface expansion [9,10]. Other Rho GTPase family members such as Rac1 [7] and the small GTPase Rsg1 [44] are involved in basal body migration.

The function of RhoA in MCCs appears to change seamlessly at basal body docking from promoting apical migration to promoting apical expansion and organizing the forming actin network. IP3R inhibition or expression of dominant-negative RhoA did not interfere with basal body docking and did not hamper long-term apical surface expansion, suggesting that regulation of RhoA activity by the IP_3_-calcium pathway is limited to post-docking stages. RhoA activity at rootlets persists at least until *Xenopus* stage 26, but it is no longer detected at stage 30. This coincides with the formation of the sub-apical actin layer, which anchors the anterior rootlets of basal bodies to the apical actin layer [18]. Rootlet anchoring also determines the direction of cilia polarization; notably, the lack of RhoA activity prevents the polarization of cilia [12]. The process of anchoring involves the ERM protein Ezrin. ERM proteins are effectors of RhoA through phosphorylation by ROCK, but can also act as negative regulators of RhoA when active [45]. Possibly, RhoA is turned off specifically after the polarization and anchoring of cilia through negative feedback by Ezrin. 

In conclusion, our findings reveal that calcium signals control the development of the apical actin network in MCCs. Stochastic calcium transients activate myosin, causing contraction of cortical and apical actin, which promotes actin network assembly from smaller actin structures (Figure 7I). Furthermore, calcium signaling activates centriolar RhoA, which is required for apical actin assembly. As development progresses, the incremental effects of individual calcium bursts accumulate over time and advance the maturation of the apical actin network. 

## 4. Materials and Methods

**Animal husbandry and microinjection.** All experiments were approved by the local authorities (Aktenzeichen 35-9185.81/G-22/043; Regierungspräsidium, Freiburg, Germany). *Xenopus laevis* embryos were obtained by in vitro fertilization [46], and staged according to [28]. Microinjections were made into the two ventral blastomeres at the four- to eight-cell stage to target the epidermis. Each blastomere was injected with 10 nL of a solution containing purified mRNAs and/or plasmid DNA, using a time- and pressure-triggered microinjection system (IM300, Narishige International Limited, Willow Business Park, Willow Way, London SE26 4QP, UK). Embryos were cultured at 12–20 °C in 0.3× Marc’s Modified Ringer solution (MMR, 30 mM NaCl, 600 µM KCl, 300 µM MgSO_4_, 600 µM CalCl_2_, 1.5 mM HEPES, pH 7.5) until they reach the desired developmental stages. The following amounts per embryo were injected: 100 pg RFP/emiRFP670-*centrin* mRNA, 150 pg GFP/mCherry-*clamp* mRNA, 100 pg GFP-rGBD mRNA, 120 pg GCaMP6s mRNA, 120 pg RFP-UtrophinCH mRNA, 250 pg BACCS2 mRNA, 300 pg dOrai mRNA, 100 pg EMTB-3xGFP mRNA, 30 pg plasmid DNA.

**Cloning.** For expression in *Xenopus* embryos, *Xenopus centrin* and *clamp,* as well as human UtrophinCH, were subcloned into pVF10 (a modified pXT7) vectors with GFP, RFP, mCherry, or emiRFP670 [47] at the N-terminus. GCaMP6s was a gift from Douglas Kim & GENIE Project (Addgene plasmid #40753, Addgene, Watertown, MA, USA) and was cloned into pCS2. BACCS2 and dOrai were a gift from Takao Nakata (Addgene plasmid #72893, Addgene, Watertown, MA, USA) and were cloned by PCR into pVF10. pCS2+GFP-rGBD was a kind gift of John Wallingford. pCS2+EMTB-3xGFP was a kind gift of Paris Skourides. Plasmids were linearized and used as templates for mRNA synthesis. In vitro transcription was performed using mMessage Machine kits (Life Technologies GmbH, Darmstadt, Germany); transcripts were purified using the Monarch RNA Cleanup kit (New England Biolabs GmbH, Frankfurt am Main, Germany).

Human RhoA with mutations (CA: F25N+Q63L; DN: T19N+F25N) was cloned into a pCS2 vector containing the multiciliated cell-specific alpha-tubulin promoter and an N-terminal GFP (pCS2.*tuba1a*:GFP-MCS). 

**Drug treatments. ***Xenopus laevis* embryos were removed from their vitelline membrane at stages 18–19 and placed in 0.3× MMR containing 50 µg/mL Gentamycin and a drug. Embryos were incubated in the drug until fixation at stages 30–32. Drug stocks were prepared in dimethyl sulfoxide (DMSO). For treatment control, embryos were incubated with a concentration of DMSO corresponding to the highest concentration of DMSO in a treated sample (Table 1).

**Confocal microscopy.** Embryos were fixed at room temperature for 40 min to 1 h with MEMFA (1 M MOPS, 2 mM EGTA, 1 mM MgSO_4_, 4% formaldehyde), followed by three washes with PBST (1× PBS, 0.1% Tween 20). For F-actin staining, Alexa Fluor 647-Phalloidin (A22287, Thermo Fisher Scientific, Life Technologies GmbH, Darmstadt, Germany) was used at a 1:300 dilution. Confocal fluorescence microscopy was performed at room temperature on Zeiss LSM880, LSM980 or Cell Discoverer 7 microscopes with AiryScan (Carl Zeiss Microscopy Deutschland GmbH, Oberkochen, Germany), using 50–63× objectives. Images were analyzed and prepared for presentation with Fiji (v1.53t) [48].

**Optogenetic induction of Calcium influx.** Embryos were injected with dOrai, BACCS2, GCaMP6s, and RFP-UtrophinCH mRNA at 4-cell stage and cultured until stages 21–22. They were then removed from the vitelline envelope using forceps and placed in a cylindrical chamber made from a cut PCR tube in a glass bottom dish (Ibidi GmbH, Gräfeling, Germany) for imaging. Imaging took place at a Zeiss LSM880 using imaging mode Airyscan Fast, 561 and 488 nm lasers, and a Plan-Apochromat 63×/1.40 oil-immersion objective. We recorded Z-stacks of single MCCs (about 20 slices, spacing 0.17 µm, centered slightly below the apical actin) every 12 s for 16 cycles. Activation of BACCS was achieved using the 488 nm laser line in Bleaching mode. MCCs in which light stimulation failed to elicit a calcium response served as controls during analysis.

For NMII inhibition during Calcium imaging, para-amino-Blebbistatin was used because of the phototoxicity and sensitivity of unmodified (−)-Blebbistatin. Para-amino-Blebbistatin or SMIFH2 was applied to embryos approximately 30 min before imaging.

Repeated illumination of embryos overnight was performed using an illumination device (optoWELL24, opto biolabs GmbH, Freiburg im Breisgau, Germany). Embryos were illuminated for 10s every 4 min at 30% light intensity with 450 nm blue light.

**Actin network analysis.** For actin network analysis, actin was visualized either by expression of RFP-UtrCH or staining with Alexa Fluor 647-Phalloidin. Confocal z-stack image series of MCCs were obtained in different regions of the embryo. For quantification in Fiji2 (v1.53t) [48], maximum intensity projections were generated and cropped along the cell border (marked by cortical actin) using “Clear outside”. The images were contrast enhanced to a saturation of 0.3%, before applying “Make Binary” and “Skeletonize”. The outer border representing the cortical actin was cropped again in the skeletonized image. The remaining apical actin skeleton was analyzed by the “Analyze Skeleton (2D/3D)” plugin. This method was inspired by and modified from [22].

We show the combined length of all actin skeletons in a cell as ‘Apical Network Length’, and as a measure of coherence the number of skeletons and the average number of branches per skeleton. A lower number of skeletons and a higher number of branches per skeleton indicate a more coherent/less fragmented actin network. Many of our treatments resulted in a moderate but significant increase in the apical cell surface area; however, we expect the length of the actin network and the number of skeletons to be proportional to that area. Therefore, we present these two parameters normalized to the apical cell surface area.

Characterization of microridges by scanning electron microscopy was performed as described previously [18].

**Ciliary rootlet polarization measurement.** Polarization of ciliary rootlets was measured in the circular standard deviation of rootlet angles, as was previously described [18].

**Immunofluorescence.** Embryos were fixed at room temperature for 40 min to 1 h with MEMFA (1 M MOPS, 2 mM EGTA, 1 mM MgSO4, 4% formaldehyde) and washed with PBST. After blocking for 1 h in PBST with 10% FBS and 5% DMSO, the embryos were incubated with mouse anti-acetylated tubulin antibody (1:1000; T6793, Sigma-Aldrich Chemie GmbH, Taufkirchen, Germany) or mouse anti-alpha-tubulin antibody (1:1000; #3873, Cell Signaling Technology, Inc., Danvers, MA, USA) overnight at 4°C. Embryos were washed with PBST and incubated with secondary anti-mouse IgG antibodies labeled with Alexa Fluor 488 (A11029, Thermo Fisher Scientific, Life Technologies GmbH, Darmstadt, Germany) or Cy3 (715-165-150, Jackson ImmunoResearch Europe Ltd., Ely, UK).

**RhoA activation analysis.** As active RhoA is localized to ciliary attachments, RFP-clamp and emiRFP670-centrin were used to highlight rootlets and basal bodies. Active RhoA was visualized by an eGFP-rhotekin G protein binding domain (rGBD)-based sensor [29]. At stages 23–24, Confocal z-stack images (objective: 50×–63× magnification) of all fluorophores were obtained. Images were analyzed with Fiji (v 1.53t). Maximum intensity projections were generated and contrast enhanced to a saturation of 0.35%. In the RFP-clamp channel, the cell was cropped and a binary mask applied, using a threshold that removes the background. The area of the rootlet was cropped using the “Analyze Particles…” tool and added to the ROI manager. To combine the area of all rootlets to one selection, the “OR (combine)” option was used. To measure the area of the cell excluding the area of the rootlets, both ROIs (cell border and area of the Rootles) were combined using the “XOR” option. The mean grey value of the eGFP-rGBD channel was measured in the previously established ROIs.

**Ciliary gliding assay.** The ciliary gliding assay was adopted from [30]. Embryos at stages 30–32 were anaesthetized in 0.02% MS-222 and placed on a 1% agarose gel. They were recorded using a Leica MZ10F Stereoscope (Leica Microsystems GmbH, Wetzlar, Germany) for 20 s with one image being taken per second. The movement of the embryos was tracked using the Fiji (v 1.53t) plugin MTrackJ (https://github.com/ImageScience).

**Statistical analysis.** Prism 9 and 10 (GraphPad Software LLC., Boston, MA, USA) were used for statistical tests and creation of graphs. The Mann–Whitney Test was used for comparisons between two unpaired samples. Wilcoxon’s matched pairs signed rank test was used for comparisons between paired samples (Figure 2B–D and Appendix A). Two-way ANOVA followed by Tukey’s multiple comparisons test was used in Figure 2E, Figure 7 and Appendix A. For Figure 5D (ciliary gliding assay), the Kruskal–Wallis test was used, followed by Dunn’s multiple comparisons test.

## Figures and Tables

**Figure 2 ijms-26-02507-f002:**
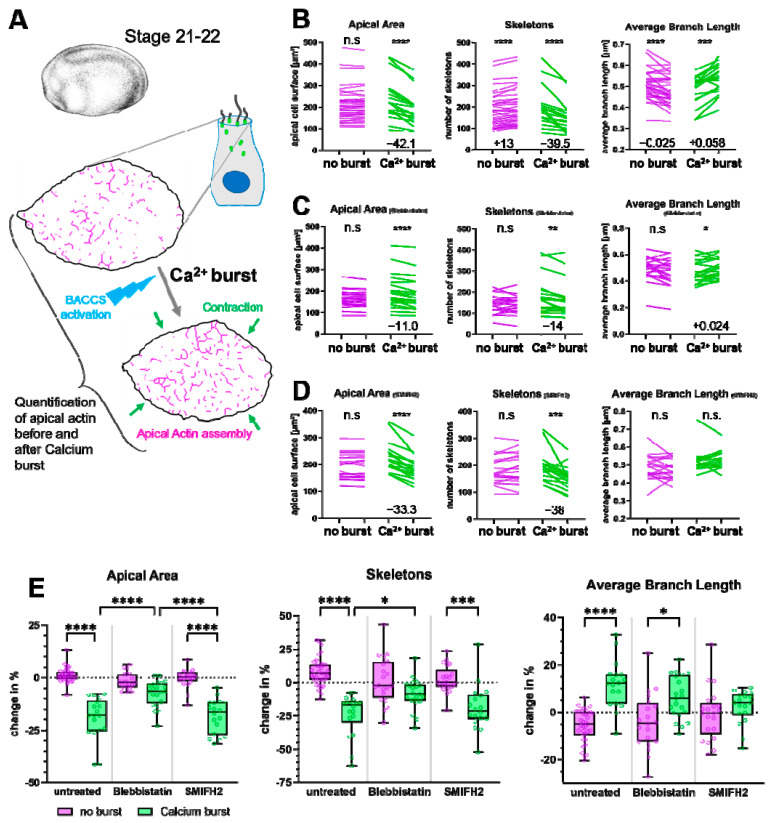
Actin network assembly in response to calcium bursts in MCCs: (**A**) Scheme: At stages 21–22, MCCs expressing BACCS components were recorded for 180 s. BACCS was activated to induce a calcium burst, resulting in apical contraction (green arrows) Apical area and the apical actin network were quantified and compared between a pre- and a post-burst timepoint. *Xenopus* illustrations © Natalya Zahn (2022) [28]. (**B**–**D**) Changes to the apical area and the actin network (number of skeletons, average branch length) in MCCs with and without calcium burst. (**B**) Untreated embryos, (**C**) Blebbistatin, (**D**) SMIFH2. Cells were recorded for 180 s and a calcium burst was induced after 12 s. Magenta lines show values for individual cells without a calcium burst, green lines for those with an induced calcium burst; end-points indicate the pre- and post-burst values. Statistical test: Wilcoxon matched pairs signed rank test. * *p* < 0.05; ** *p* < 0.01; *** *p* < 0.001; **** *p* < 0.0001; n.s. not significant. For significant changes, the median change is included in the graph. Mean and SD values are provided in Appendix A. (**E**) Comparison of changes to the apical area and the actin network (number of skeletons, average branch length) between MCCs with and without calcium activity both for untreated, Blebbistatin-treated, and SMIFH2-treated embryos. Graphs show the changes presented in (**B**–**D**) in percent. Each data point represents an individual cell. Box and whiskers indicate quartiles. Dashed line indicates 0% (no change). Averages and SD: Apical Area (Untreated, no burst: +1.2 ± 3.5%; calcium burst: −19.3 ± 10.0%; Blebbistatin, no burst: −1.7 ± 3.9%; calcium burst: −7.5 ± 6.1%; SMIFH2, no burst: −0.3 ± 4.4%; calcium burst: −18.3 ± 8.1%). Skeleton number (Untreated, no burst: +8.1 ± 10.5%; calcium burst: −23.9 ± 15.8%; Blebbistatin, no burst: +1.6 ± 17.5%; calcium burst: −8.7 ± 11.5%; SMIFH2, no burst: +2.2 ± 10.1%; calcium burst: −18.9 ± 17.1%). Average branch length (Untreated, no burst: −5.7 ± 6.8%; calcium burst: +11.7 ± 10.0%; Blebbistatin, no burst: −4.1 ± 11.8%; calcium burst: +6.0 ± 9.0%; SMIFH2, no burst: −1.2 ± 11.0%; calcium burst: +2.2 ± 7.1%). Statistical test: 2-way ANOVA + Tukey’s multiple comparisons test. * *p* < 0.05; ** *p* < 0.01; *** *p* < 0.001; **** *p* < 0.0001; non-significant comparisons are not shown; 53 cells from 30 untreated embryos; 41 cells from 14 Blebbistatin-treated embryos; 39 cells from 18 SMIFH2-treated embryos.

**Figure 3 ijms-26-02507-f003:**
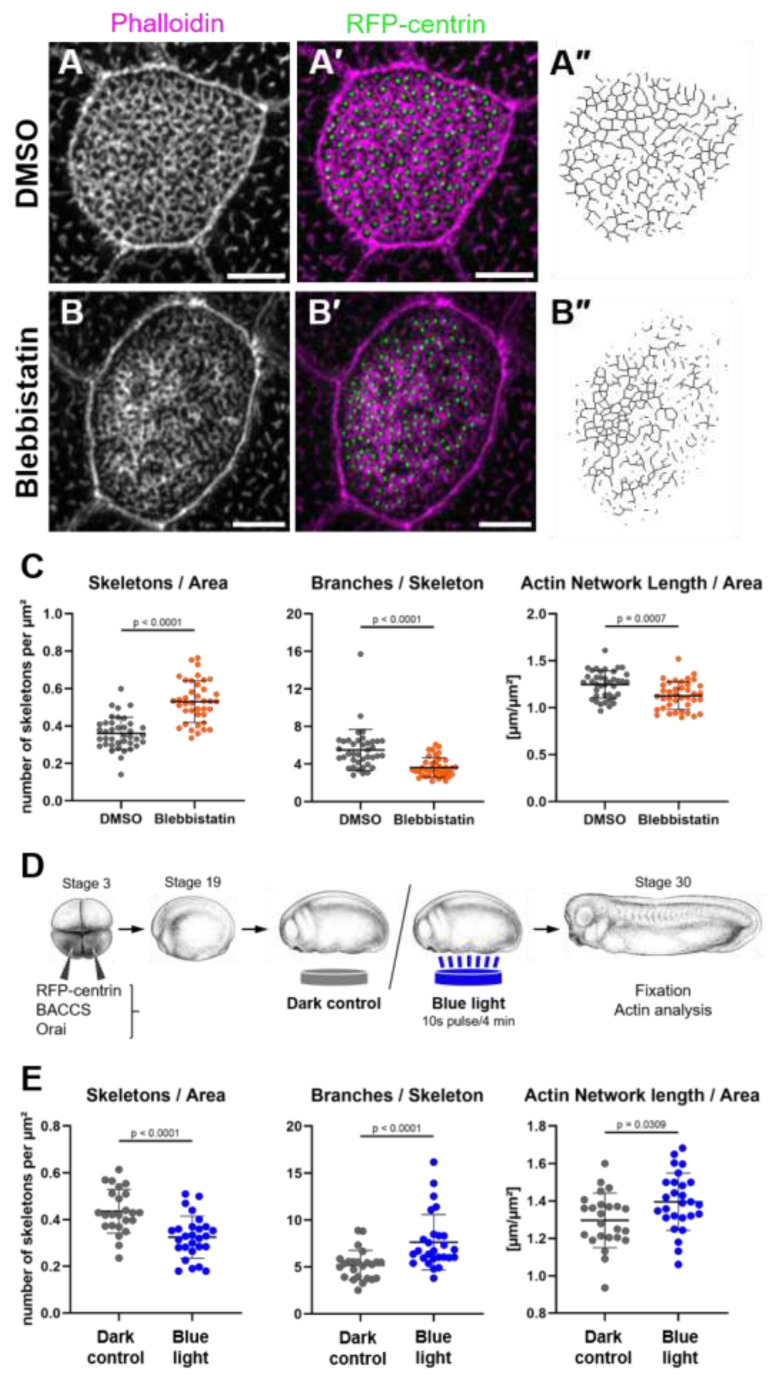
Non-muscle myosin II and recurring calcium bursts drive apical actin network development: (**A**–**C**) Maximum intensity projections of apical actin in MCCs of embryos treated with (**A**) DMSO or (**B**) 50 µM Blebbistatin from stage 19 to 30. (**A’**,**B’**) show actin labeled by phalloidin in magenta, and GFP-centrin-labeled basal bodies in green. Scale bars 5 µm. (**A’’**,**B’’**) show the skeletonized actin network. (**C**) Quantification of the apical actin network of MCCs in embryos treated with DMSO or Blebbistatin. Number of skeletons (separate actin structures) per µm^2^ (DMSO 0.36 ± 0.09; Blebbistatin 0.53 ± 0.11). Average branches per skeleton (DMSO 5.5 ± 2.2; Blebbistatin 3.6 ± 1.0). Length of the apical actin network normalized to surface area (DMSO 1.25 ± 0.15 µm^−1^; Blebbistatin 1.13 ± 0.15 µm^−1^). Error bars show mean and SD; 39 cells from 12 to 13 embryos analyzed per condition. Statistical test: Mann–Whitney Test (**D**,**E**) Effect of repeated induction of calcium bursts. (**D**) Experimental setup: Embryos expressing BACCS and Orai were grown in darkness or with 10 s pulses of 450 nm light every 4 min from stage 19 to 30. *Xenopus* illustrations © Natalya Zahn (2022) [28]. (**E**) Quantification of MCC apical actin: Number of skeletons per µm^2^ (Dark 0.44 ± 0.09; Blue 0.33 ± 0.09). Average branches per skeleton (Dark 5.2 ± 1.6; Blue 7.7 ± 3.0). Length of the apical actin network normalized to surface area (Dark 1.30 ± 0.15 µm^−1^; Blue 1.40 ± 0.15 µm^−1^). Error bars show mean and SD; 24–27 cells from 8 to 9 embryos analyzed per condition. Statistical test: Mann–Whitney Test.

**Figure 4 ijms-26-02507-f004:**
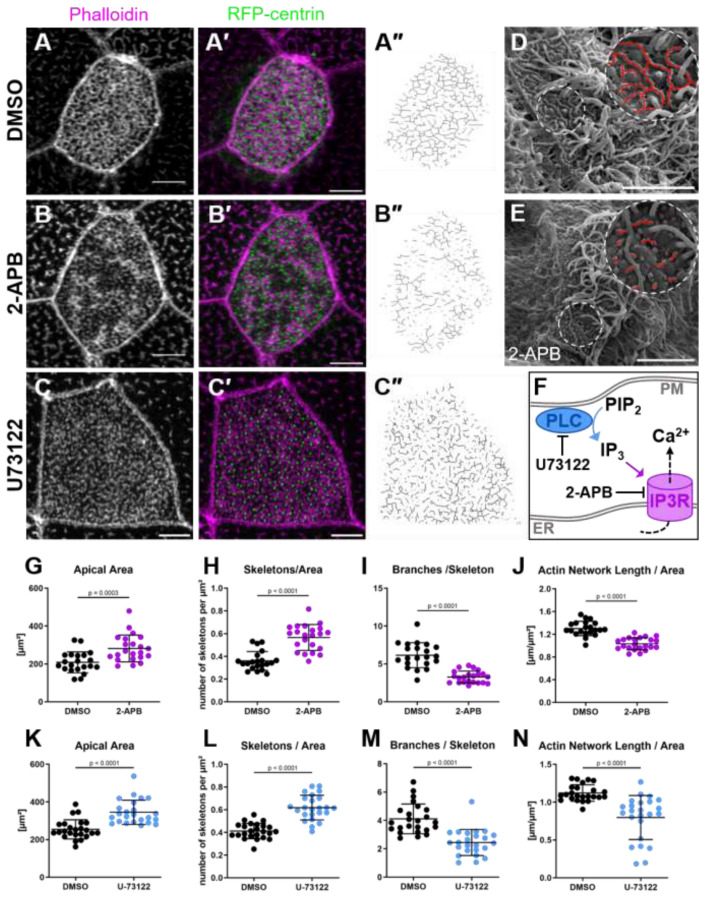
Inhibition of IP3R and PLC leads to truncation of the apical actin network and microridges: (**A**–**C**) Maximum intensity projections of apical actin in MCCs of embryos treated with (**A**) DMSO, (**B**) 2-APB, or (**C**) U73122 from stage 19 to 30. (**A’**–**C’**) show actin labeled by phalloidin in magenta, and RFP-centrin-labeled basal bodies in green. (**A’’**–**C’’**) show the skeletonized actin network. Scale bars 5 µm. (**D**,**E**) Scanning electron micrographs of MCCs on the surfaces of embryos treated with (**D**) DMSO or (**E**) 2-APB. Microridges were manually labeled red in magnified insets (dashed circles). Scale bar 5 µm. (**F**) Scheme illustrating the interactions of small molecule inhibitors 2-APB and U73122 with the IP_3_-calcium pathway. IP_3_, inositol 1,4,5-triphosphate; IP3R, inositol triphosphate receptor; PIP_2_, phosphatidylinositol 4,5-bisphosphate; PLC, phospholipase C; PM, plasma membrane; ER, endoplasmic reticulum membrane. (**G**–**J**) Quantification of apical surface area and the apical actin network of MCCs in embryos treated with DMSO or 2-APB. (**G**) Apical surface area (DMSO: 208.5 ± 55.4 µm^2^; 2-APB: 282.4 ± 71.0 µm^2^). (**H**) Number of skeletons (separate actin structures) per µm^2^ (DMSO: 0.36 ± 0.08; 2-APB: 0.57 ± 0.11). (**I**) Average branches per skeleton (DMSO: 6.2 ± 1.7; 2-APB: 3.3 ± 0.8). (**J**) Length of the apical actin network normalized to surface area (DMSO: 1.30 ± 0.13 µm^−1^; 2-APB: 1.03 ± 0.11 µm^−1^). (**K**–**N**) Quantification of apical surface area and the apical actin network of MCCs in embryos treated with DMSO or U73122. (**K**) Apical surface area (DMSO 254.3 ± 51.2 µm^2^; U73122 344.9 ± 64.5 µm^2^). (**L**) Number of skeletons (separate actin structures) per µm^2^ (DMSO 0.41 ± 0.07; U73122 0.62 ± 0.11). (**M**) Average branches per skeleton (DMSO 4.1 ± 1.1; U73122 2.4 ± 0.9). (**N**) Length of the apical actin network normalized to surface area (DMSO 1.12 ± 0.11 µm^−1^; U73122 0.80 ± 0.29 µm^−1^). Error bars show mean and SD; 21–24 cells from 6 to 8 embryos analyzed per condition. Statistical test: Mann–Whitney Test.

**Figure 5 ijms-26-02507-f005:**
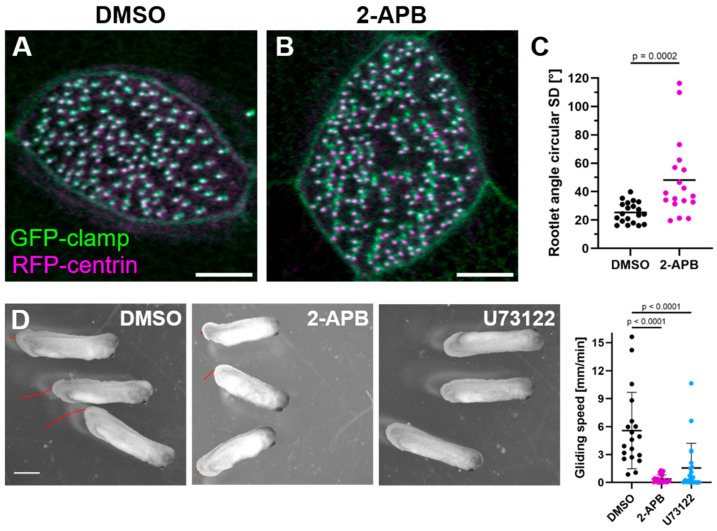
(**A**–**C**) Polarization of ciliary rootlets in MCCs treated with (**A**) DMSO and (**B**) 2-APB. Ciliary rootlets labeled by GFP-clamp are in green, and basal bodies labeled by RFP-centrin are in magenta (Maximum intensity projections). Scale bar 5 µm. (**C**) Rootlet angle circular standard deviation in MCCs (DMSO 25.4° mean, 24.3° median; 2-APB 48.2° mean, 38.0° median). Statistical test: Mann–Whitney test (**D**) Ciliary Gliding Assay. Stage 30 embryos treated with DMSO, 2-APB, or U73122 starting stage 19. Red traces indicate distance moved within 20 s. Scale bar 1 mm. In the graph, bars indicate the median (DMSO: 4.61 mm/min; 2-APB: 0.21 mm/min; 0.49 mm/min); 19–21 embryos analyzed in two independent experiments. Statistical Test: Kruskal–Wallis Test + Dunn’s multiple comparisons test.

**Figure 6 ijms-26-02507-f006:**
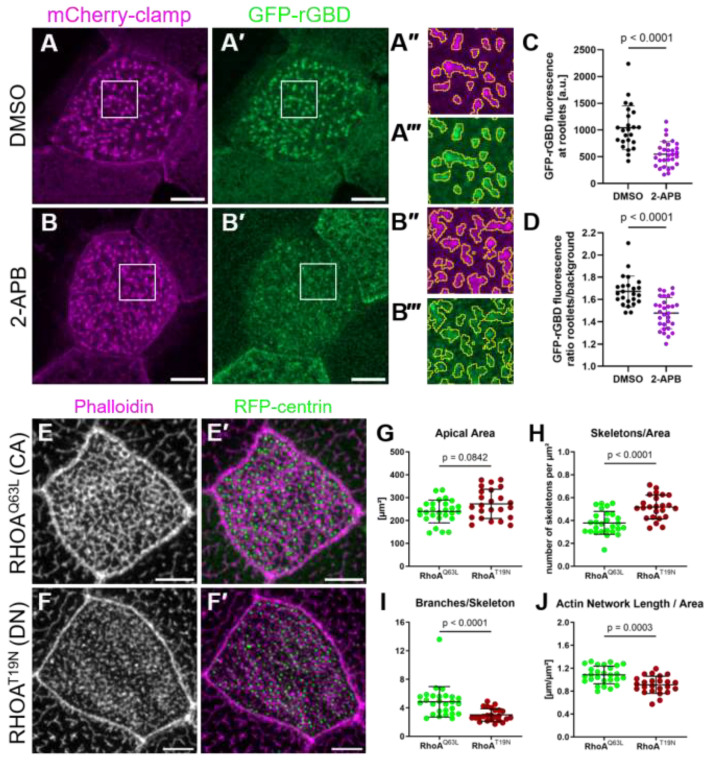
Inhibition of IP3R disrupts the apical actin network by interfering with local activation of RhoA: (**A**–**D**) Active RhoA (GFP-rGBD, green, **A’**,**B’**) localizes to the anterior rootlet (mCherry-clamp, magenta) of basal bodies in MCCs of control embryos treated with DMSO from stage 19 to 24 (Maximum intensity projections). Boxed areas are magnified in (**A’’**,**A’’’**,**B’’**,**B’’’**). Yellow outlines in magnified areas indicate basal body rootlets. Scale bars 5 µm. (**C**) Graph comparing the average fluorescence intensity (a.u.) within the yellow outlines in (**A’’’**,**B’’’**) between DMSO and 2-APB treatments. (DMSO: 1043 ± 410; 2-APB: 548 ± 237). (**D**) Rootlet/cell background ratio of GFP-rGBD fluorescence compared between DMSO and 2-APB treatments. (DMSO: 1.67 ± 0.14; 2-APB: 1.48 ± 0.14); 24–30 cells from 8 embryos analyzed per condition. Statistical test: Mann–Whitney Test. (**E**,**F**) Maximum intensity projections of apical actin in MCCs expressing (**E**) GFP-RHOA^Q63L^ or (**F**) GFP-RHOA^T19N^. (**E’**,**F’**) show actin labeled by phalloidin in magenta, and RFP-centrin-labeled basal bodies in green. Scale bars 5 µm. (**G**–**J**) Quantification of the apical surface area and the apical actin network of MCCs expressing GFP-RHOA^Q63L^ or GFP-RHOA^T19N^. (**G**) Apical area (RHOA^Q63L^ 239.5 ± 49.7 µm^2^; RHOA^T19N^ 272.83 ± 64.1 µm^2^). (**H**) Number of skeletons (separate actin structures) per µm^2^ (RHOA^Q63L^ 0.38 ± 0.10; RHOA^T19N^ 0.52 ± 0.10). (**I**) Average branches per skeleton (RHOA^Q63L^ 4.9 ± 2.1; RHOA^T19N^ 3.0 ± 0.9). (**J**) Length of the apical actin network normalized to surface area (RHOA^Q63L^ 1.09 ± 0.15 µm^−1^; RHOA^T19N^ 0.91 ± 0.15 µm^−1^). Error bars show mean and SD; 24–27 cells from 8 embryos analyzed per condition. Statistical test: Mann–Whitney Test.

**Figure 7 ijms-26-02507-f007:**
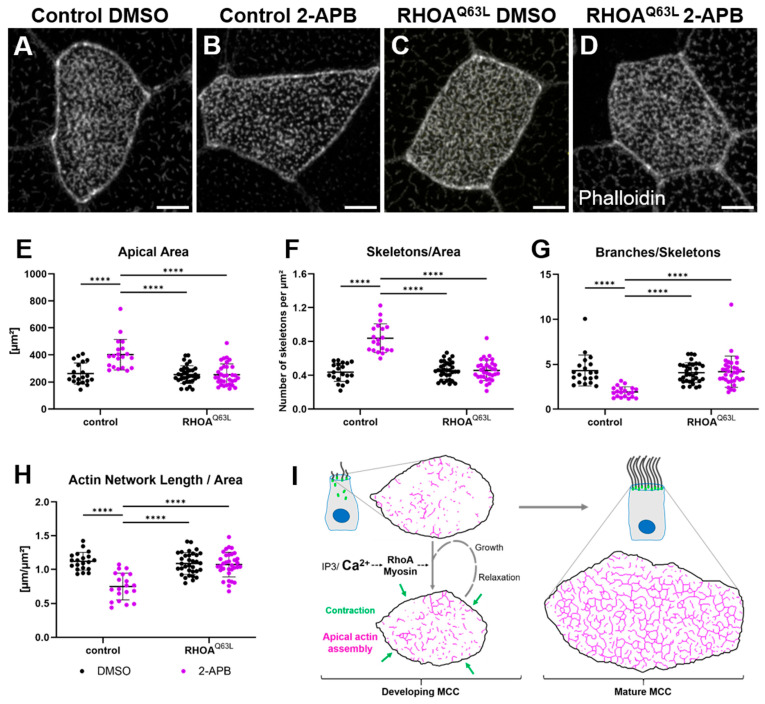
Apical actin truncation due to IP3R inhibition is rescued by constitutively active RhoA: (**A**–**H**) Maximum intensity projections of apical actin and quantification thereof in MCCs treated with DMSO or 2-APB and expressing constitutively active RHOA^Q63L^. Scale bars 5 µm. (**E**) Apical surface area (DMSO control: 263.3 ± 78.2 µm^2^; 2-APB control: 402.1 ± 112.2 µm^2^; DMSO RHOA^Q63L^: 255.5 ± 65.6 µm^2^; 2-APB RHOA^Q63L^: 254.5 ± 79.7 µm^2^). (**F**) Number of skeletons (separate actin structures) per µm^2^ (DMSO control: 0.44 ± 0.11; 2-APB control: 0.84 ± 0.17; DMSO RHOA^Q63L^: 0.46 ± 0.10; 2-APB RHOA^Q63L^: 0.46 ± 0.12). (**G**) Average branches per skeleton (DMSO control: 4.3 ± 1.7; 2-APB control: 1.9 ± 0.6; DMSO RHOA^Q63L^: 4.1 ± 1.1; 2-APB RHOA^Q63L^: 4.2 ± 1.7). (**H**) Length of the apical actin network normalized to surface area (DMSO control: 1.12 ± 0.13 µm^−1^; 2-APB control: 0.75 ± 0.20 µm^−1^; DMSO RHOA^Q63L^: 1.09 ± 0.16 µm^−1^; 2-APB RHOA^Q63L^: 1.07 ± 0.18 µm^−1^). Error bars show mean and SD; 31–51 cells from 10 to 16 embryos analyzed per condition. Statistical test: 2-way ANOVA + Tukey’s multiple comparisons test. **** *p* < 0.0001. (**I**) Model depicting the role of calcium, myosin, and RhoA in the maturation of the apical actin network. Green arrows symbolize the apical contraction following a calcium burst. Apical actin is depicted skeletonized in pink.

**Table 1 ijms-26-02507-t001:** Drugs used in this study.

Drug	Supplier	Stock Concentration	Final Concentration (DMSO conc.)
2-Aminoethylborate (2-APB)	Tocris (Bio-Techne GmbH, Wiesbaden, Germany)	20 mM	15 µM (0.075%)
U73122	Tocris (Bio-Techne GmbH, Wiesbaden, Germany)	5 mM	0.5 µM (0.01%)
Y27632	Tocris (Bio-Techne GmbH, Wiesbaden, Germany)	50 mM	20 µM (0.04%)
(−)-Blebbistatin	Sigma-Aldrich Chemie GmbH, Taufkirchen, Germany	25 mM	50 µM (0.2%)
para-amino-Blebbistatin	Cayman Chemical, Ann Arbor, MI, USA	100 mM	100 µM (0.1%)
SMIFH2	Tocris (Bio-Techne GmbH, Wiesbaden, Germany)	20 mM	10 µM (0.05%)
YM 58483	Tocris (Bio-Techne GmbH, Wiesbaden, Germany)	20 mM	40 µM (0.2%)

## Data Availability

The raw data supporting the conclusions of this article will be made available by the authors on request.

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
