# Peer review of "Spontaneous Calcium Bursts Organize the Apical Actin Cytoskeleton of Multiciliated Cells"

_ijms, 2025, doi:10.3390/ijms26062507_

Round 1

Reviewer 1 Report

Comments and Suggestions for Authors

The manuscript by Wiegel et al. investigates formation of motile cilia using Xenopus embryo epidermis as a multiciliated cell model. Using an elegant optogenetic approach, the authors found that calcium signaling promotes assembly of apical actin networks required for maturation of the multiciliated cells. The authors demonstrate that this process is RhoA and non-muscle myosin dependent. Given the involvement of motile cilia in numerous physiological processes as well as several disease conditions, the study is interesting and important. The manuscript is very well written, and the findings are nicely presented. I appreciate the authors’ effort put into the schematic drawings, which are extremely helpful and beautifully illustrated. There are several issues noted and need to be addressed before the manuscript can be accepted.

Major comment:

The authors use SMIFH2 as a “small molecule inhibitor of FH2 domain” of formins. However, this drug has off-target effect: it potently inhibits myosin family of motors (including non-muscle myosin II). The reference is PMID: 33589498. This has to be reflected in the manuscript especially in light that myosin inhibition with blebbistatin plays an important role in the described process. Given that moderate concentrations of SMIFH2 were used, and the effects of SMIFH2 and blebbistatin were quantitatively different, it is possible to provide a statement like this in the text without the need for alternative experiments.

Minor comments:

  1. Page 2: A brief introduction (one-two sentences) describing both calcium indicator GCaMP6s and BACCS system and its components (e.g. Orai) would really help a general reader.
  2. Page 4 line 106: Is the word “burst” missing between “calcium” and “(Fig. 2A)”?
  3. Page 15 line 422: Can the formulation or the source of MMR solution be provided?
  4. Page 16 line 436: What is the role of F25N mutation in human RhoA constructs?
  5. Page 16 line 440: What was the concentration of gentamycin used in the experiments?
  6. Page 17 line 498: Was any mounting medium used for the embryo immunofluorescence experiments?
  7. Page 17 line 500: Replace “was” with “were”.
  8. Page 17 line 502: What is “NF St.23-24”?
  9. Figure 1A: The panel is labeled as “Phalloidin”, which cannot be right for live-cell imaging. The legend says: “RFP-utrophin”, is this correct and should replace the incorrect label?
  10. What are the scale bars on Fig. 1A, 4D and E, 5D, S5G-J, S6A-B’?
  11. Number of cells analyzed in Fig. S2?
  12. Please rephrase Fig. S2A-C legend to correctly describe what the lines indicate (e.g., something similar to the Fig. 2B-D legend).

Reviewer 2 Report

Comments and Suggestions for Authors

In the current manuscript investigate how multiciliated cells (MCCs) control the development of the apical actin network and microridges. Using xenopus embryos as models, they observe stochastic calcium bursts in MCCs during ciliogenesis. They reported a novel role for calcium during MCC development, driving apical contractions and actin remodeling. Using optogenetics, they propose that repetitive and transient calcium bursts together with consequent apical contractions supports the extension and increased interconnectivity of the apical actin network in individual cells.

This is a well conducted study and a well written manuscript, whose conclusions are of the interest of a broad scientific community. I recommend the publication of this manuscript in its current form in IJMS.

Author Response

Comments: 

In the current manuscript investigate how multiciliated cells (MCCs) control the development of the apical actin network and microridges. Using xenopus embryos as models, they observe stochastic calcium bursts in MCCs during ciliogenesis. They reported a novel role for calcium during MCC development, driving apical contractions and actin remodeling. Using optogenetics, they propose that repetitive and transient calcium bursts together with consequent apical contractions supports the extension and increased interconnectivity of the apical actin network in individual cells.

This is a well conducted study and a well written manuscript, whose conclusions are of the interest of a broad scientific community. I recommend the publication of this manuscript in its current form in IJMS.

Reply: We thank the reviewer for their time and their kind evaluation of our manuscript.